# Adding $^{161}$Dy-Mössbauer spectroscopy to a multitechnique investigation of magnetic transitions in a {Co$^{III}_3$Dy$^{III}_3$} Single-Molecule Toroic

Yan Peng [1,2] ✉, Jonas Braun [1,3,4], Lena Scherthan[5], Hendrik Auerbach[6], Juliusz A. Wolny[5], Michael Schulze[7], E. Ercan Alp [8], Jiyong Zhao[8], Wenli Bi [9], Lorenzo Tesi [10,11], Christopher E. Anson [1], Jani O. Moilanen [12], Dennis. E. Brown[13], Liviu F. Chibotaru [14], Wolfgang Wernsdorfer [4,7], Mauro Perfetti [10], Roberta Sessoli [10] ✉, Volker Schünemann [5] ✉ & Annie K. Powell [1,3,4] ✉

The determination of the orientations of the individual Dy$^{III}$ anisotropy axes in polynuclear complexes is challenging but crucial for the understanding of systems showing Single Molecule Magnet or Single Molecule Toroic behavior. In particular, the experimental proof of a toroidal ground state from magnetization data often remains ambiguous. Here, we report the coordination cluster [Co$^{III}_3$Dy$^{III}_3$($\mu_3$-OH)$_4$(O$_2$C-C$_6$H$_4$-p-Me)$_6$(pmide)$_3$(H$_2$O)$_3$]Cl$_2$ · 10MeCN (1) (H$_2$pmide = N-2-pyridylmethyldiethanolamine) which crystallizes with three-fold symmetry and contains an equilateral Dy$^{III}_3$ triangle surrounded by a triangle of diamagnetic Co$^{III}$ ions. We also report a multi-technique investigation of its toroidal magnetic spin structure, including $^{161}$Dy Synchrotron Mössbauer Spectroscopy which shows an abrupt transition from a non-magnetic to a magnetic state. The experimental orientations of the individual Dy$^{III}$ magnetic axes were assessed using torque magnetometry and micro-SQUID measurements and both experiments converged on a spin structure that is in very good agreement with ab initio calculations. Such a multi-technique approach, including $^{161}$Dy Synchrotron Mössbauer Spectroscopy, provides a roadmap for the unambiguous identification of such toroidal states.

For anisotropic spin cluster systems, the phenomenon of Single Molecule Magnet (SMM) behavior, i.e., slow relaxation of the magnetization, can be observed[1]. Particularly, mononuclear lanthanide-based systems can be tailored to be high-performance SMMs using ligand fields that maximize magnetic anisotropy[2–4]. The control of individual anisotropies in polynuclear lanthanide clusters is more challenging, although specific arrangements of anisotropy axes can lead to exotic spin structures. Here, spin structure refers to the orientation of the Dy

anisotropy axes in one molecular entity. A special case observed for some polynuclear SMMs is the presence of an essentially non-magnetic ground state arising from the non-collinear antiferromagnetic arrangement of Ising spins to give a toroidal moment. Such toroidal moments are anapolar and can be visualized as possessing a two-fold degenerate ground state of opposite spin chirality with the spins arranged like a left or right-handed propeller (axial toroidal moment) or like a left- and right-handed helical twisting of the axes (polar

toroidal moment)[5–8]. This suggests that information can be stored utilizing the handedness of the toroidal arrangement of the magnetic moments. Thus, the toroidal arrangement can be detected through analysis of the magnetization data, where the nearly non-magnetic, toroidal state persists up to a characteristic applied field and then jumps to a magnetic state when the excited magnetic state dips below the non-magnetic ground state. The SMM regime is thus accessed through the application of a magnetic field. This phenomenon was first discovered in a triangular [$Dy_3(\mu_3$-OH)$_2$($o$-van)$_3$Cl(H$_2$O)$_5$] Cl$_3$·4H$_2$O·2MeOH·0.7MeCN molecule ($o$-van = 2-hydroxy-3-methoxybenzaldehyde)[9,10]. These results, together with further measurements using high-frequency electron paramagnetic resonance (HF-EPR), magnetic far-infrared spectroscopies and cantilever torque magnetometry[11], provide a comprehensive study of the intricacies of the electronic/magnetic structure and properties of this archetypal Dy$_3$ molecule.

Here, we report a multi-technique study on a new Co$^{III}_3$Dy$^{III}_3$ coordination cluster with trigonal symmetry, which corresponds magnetically to a Dy$^{III}_3$ triangle, noting that low-spin Co$^{III}$ is diamagnetic. Magnetic measurements performed on single crystals of the cluster give information on the orientation of the principal anisotropy axes of the ground Kramers' doublets of the Dy$^{III}$ ions in the cluster. The overall picture is that the three magnetic moments are each tilted by ca. 15° out of the plane of the triangle, providing a small out-of-plane dipole moment in addition to a large toroidal one. We also used $^{161}$Dy

time-domain Synchrotron Mössbauer Spectroscopy (SMS), also referred to as Nuclear Forward Scattering (NFS), in a previously published pilot study on the mononuclear SMM [Dy(Cy$_3$PO)$_2$(H$_2$O)$_5$] Br$_3$·2(Cy$_3$PO)·2H$_2$O·2EtOH[12]. With $^{161}$Dy-SMS, the sample's magnetic properties can be sensed from the inside of the molecules, and there is no need to disturb the system under study with external magnetic fields (as in magnetometry) or with radiofrequency fields (as in electron paramagnetic resonance). Even more, since the resonant recoil-free scattering process of the Mössbauer quanta is a coherent one, the unavoidable quantum mechanical disturbance due to the measuring process is minimized.

Here, $^{161}$Dy time-domain SMS allowed the identification of the critical field at which the toroidal regime crosses over to the SMM state. This provides an independent confirmation of the results from the magnetization data. These experimental results are not only in good agreement with those obtained from ab initio calculations, but also demonstrate the effects of a strongly hydrogen-bonded chloride counterion on the spin structure of the cluster, suggesting that counterion optimization could be a hitherto unexplored chemical mean to tune the toroidal moments.

## Results and discussion
### Crystal structure analysis
The compound [Co$^{III}_3$Dy$^{III}_3$($\mu_3$−OH)$_4$(O$_2$C−C$_6$H$_4$-$p$−Me)$_6$(pmide)$_3$(H$_2$O)$_3$]Cl$_2$ · 10MeCN (**1**) (where H$_2$pmide = $N$−2-

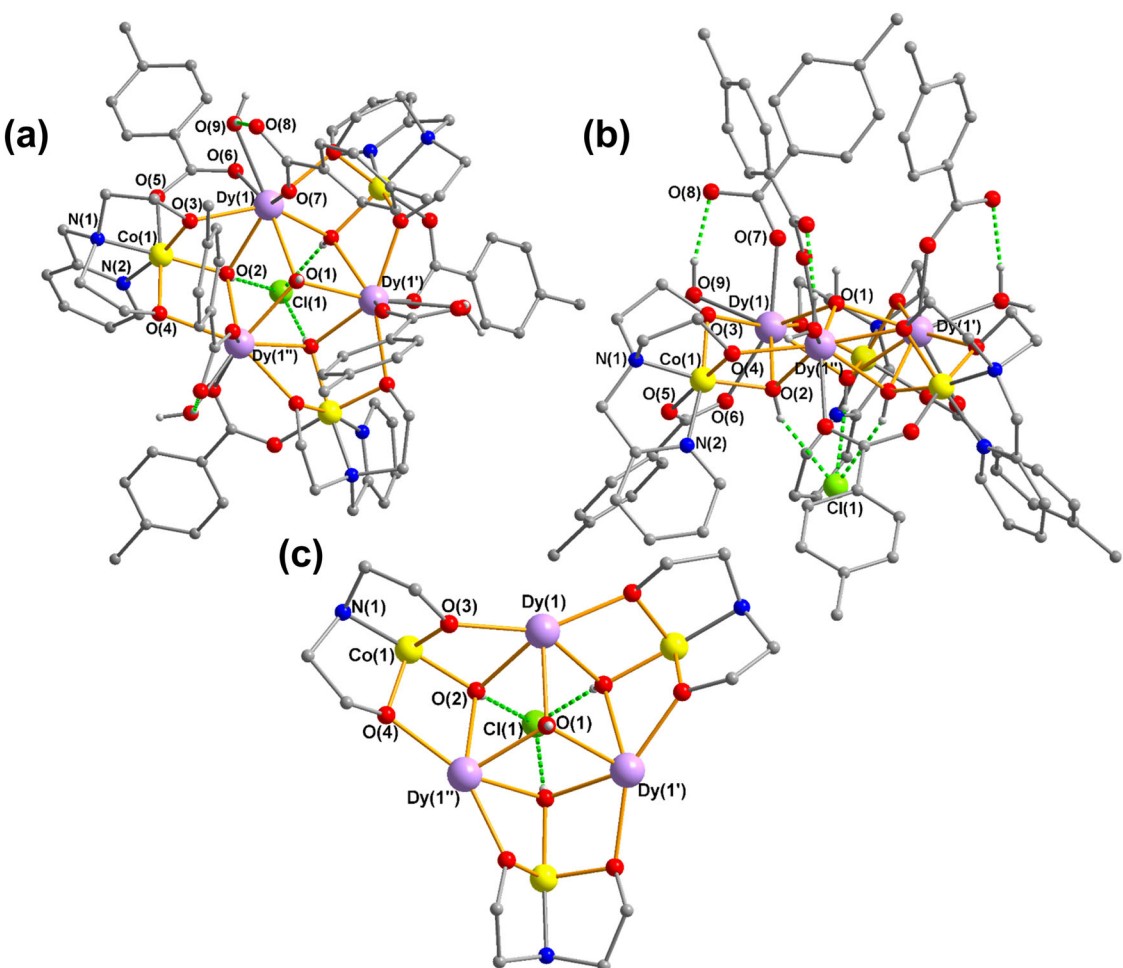

**Fig. 1 | Molecular Structure of 1. a** Molecular structure of the Co$_3$Dy$_3$ cluster together with the hydrogen-bonded chloride counteranion in **1**, viewed almost down the crystal threefold axis. **b** as (**a**), but side view showing the "shuttlecock" shape of the cluster. **c** Central core of the cluster, omitting pyridyl groups and toluate and water ligands. Dy: violet, Cl: green, O: red, H: white, N: blue, C: gray, organic H-atoms omitted for clarity; hydrogen bonds shown as green dashed lines.

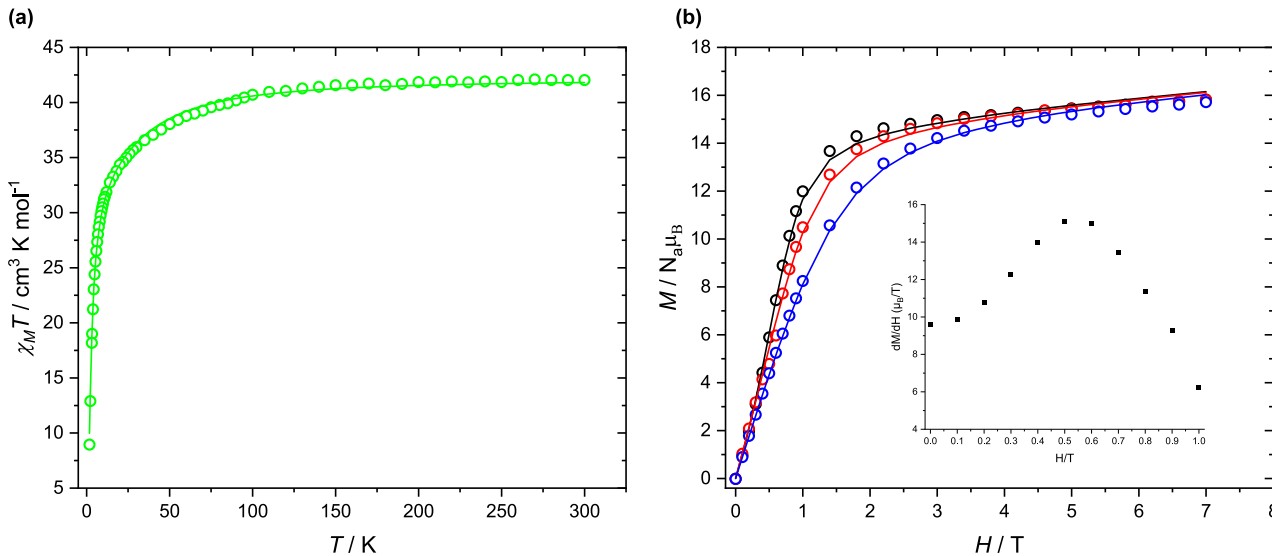

**Fig. 2 | d.c. Magnetometry Data for 1. a** Plot of χT vs T for **1** at 1000 Oe and (**b**) plots of M vs H at 2, 3 and 5 K (black, red and blue circles, respectively), solid line from fitting. Inset: First derivative of the magnetization with respect to field for **1** as a function of the applied field at 2 K; the maximum of this quantity corresponds to the position of an inflection point in the magnetization curves.

pyridylmethyldiethanolamine) crystallizes in the trigonal space group $P\bar{3}c1$ with Z = 4. Crystallographic parameters and refinement details are listed in Supplementary Table S1. The dicationic cluster (Fig. 1a, b) is based upon a triangle of three $Dy^{III}$ ions bridged by a $\mu_3$-OH ligand at its center. Each pair of $Dy^{III}$ in the triangle is then linked by a further $\mu_3$-OH to a $Co^{III}$ ion, resulting in a $\{Co_3Dy_3(\mu_3\text{-OH})_4\}$ core. Each $Co^{III}$ is chelated by a (pmide)$^{2-}$ ligand, of which the two deprotonated oxygens each bridge between the $Co^{III}$ and a $Dy^{III}$, stabilizing the core (Fig. 1c). Further bridging is provided by *syn,syn*-bridging carboxylates from three *p*-toluate ligands, while the coordination environment is completed by three monodentate toluates and three water ligands. Selected bond lengths and angles are listed in Supplementary Table S2.

Co(1) is six-coordinate, adopting an octahedral $N_2O_4$ geometry with an O atom from a $\mu_3$-OH group, one carboxylate O atom and two O atoms and two N atoms from its chelating (pmide)$^{2-}$ ligand. The deviation from idealized octahedral coordination geometry is only 0.66 as calculated using the SHAPE program[13,14]. The Co-N/O distances are in the range 1.859(3)–1.929(4) Å, consistent with low-spin $d^6$ dia-magnetic $Co^{III}$. Dy(1) is eight coordinate with Dy-O distances in the range 2.300(3)–2.482(4) Å, and SHAPE analysis reveals that its geo-metry is best described as biaugmented trigonal prismatic (BTPR-8, $C_{2v}$, 1.37).

The cluster molecule has threefold site symmetry, with a crystal-lographic $C_3$ axis passing through the central ($\mu_3$-OH) bridge O(1) and also through a chloride counteranion Cl(1). This chloride accepts three equivalent hydrogen bonds from O(2) and its symmetry-equivalents with a Dy(1)···Cl(1) distance of 4.521(2) Å. The second chloride in the structure, Cl(2), is disordered over three equivalent sites in the struc-ture and is much further from the cluster than Cl(1), with Dy(1)···Cl(2) 7.19(2) Å. The metal core in **1** can be viewed as built from two stacked equilateral triangles: one defined by the three Dy ions with a Dy···Dy distance of 3.8503(5) Å, and a larger triangle formed by the three Co sites [Co···Co, 6.2765(13) Å]. These two triangles are co-parallel by symmetry, and 1.046 Å apart.

### Characterization of polycrystalline samples
Polycrystalline samples are useful for providing a global picture of bulk magnetic and electronic properties. Here, bulk magnetic susceptibility studies provide insights into both the static and dynamic magnetism via dc and ac measurements. These measurements have been

supplemented by a direct probe of the Dy nuclei using $^{161}$Dy time-domain SMS and show how this nucleus responds to the shift from the toroidal non-magnetic ground state to a magnetic state on application of a magnetic field.

### SQUID measurements
Variable-temperature magnetization measurement of **1** performed in the range 1.8–300 K at 1000 Oe (Fig. 2a) were used to extract the magnetic susceptibility as χ = M/H. The value of χT is 42.03 cm$^3$mol$^{-1}$K at room temperature, which is in good agreement with the expected value of 42.42 cm$^3$K mol$^{-1}$ for three paramagnetic non-interacting $Dy^{III}$ ions[9]. The χT value for compound **1** decreases gradually between 300 and 30 K and then drops steeply to reach the minimum value of 8.94 cm$^3$mol$^{-1}$K at 1.8 K. The decrease of $\chi_M T$ with decreasing tem-perature can be most appropriately explained by the thermal depopulation of the excited $m_J$ sublevels of the anisotropic $Dy^{III}$ ions[15]. However, the pronounced drop at the lowest temperatures probably indicates an antiferromagnetic coupling of the $Dy^{III}$ ions, as also sup-ported by ab initio calculations (see below). The magnetization of **1** at 2 K increases upon application of an external field to a maximum of 15.57 $N_A\mu_B$ at 7 T, close to the expected value 3 × 5.23 = 15.69 $N_A\mu_B$ for a $Dy_3$ triangle[9]. An inflection point indicates the presence of a level crossing from a non-magnetic, possibly toroidal ground state, result-ing from magnetic anisotropy and intramolecular antiferromagnetic interactions, to a ferromagnetic excited state, reminiscent of the situation in the original $Dy_3$ triangle compound[9,10,16]. The characteristic field value is best revealed by plotting the first derivative of the mag-netization curve[17], showing that this level crossing in **1** occurs at a field of ~0.5 T (Fig. 2b, inset).

To investigate the dynamics of the magnetization, ac suscept-ibility studies were performed for **1**. Similar dynamic magnetic beha-vior with broad frequency-dependent peaks under zero dc field (see Supplementary Fig. S4b) as seen in previously reported $Dy_3$ clusters[9,18], suggests the presence of more than one relaxation process that cannot be satisfactorily fitted using a double Debye model.

At an applied dc field of 3000 Oe, the two processes lie in the same frequency range with heavy overlap. As a result of this overlap the Cole-Cole plots (Supplementary Fig. S8, left) were fitted using a single generalized Debye model, resulting in the relaxation times shown in Supplementary Fig. S8, right and α parameters in the range

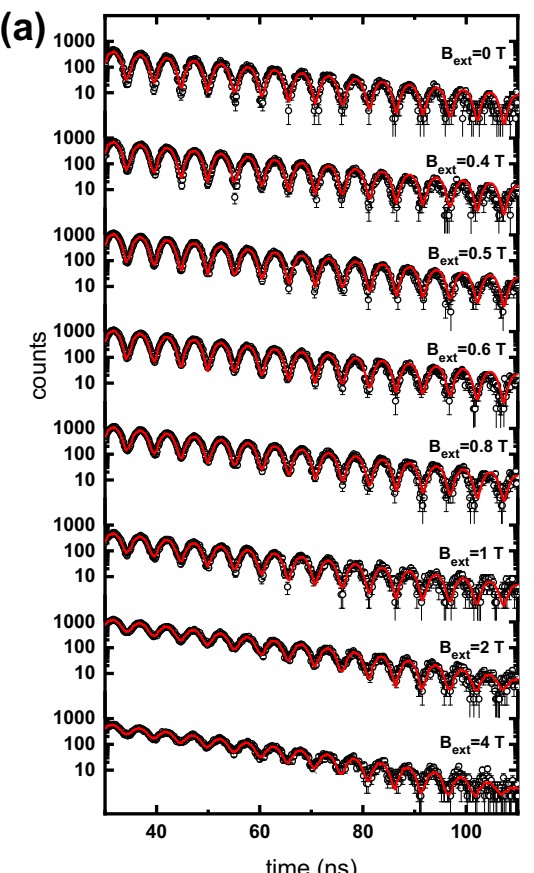

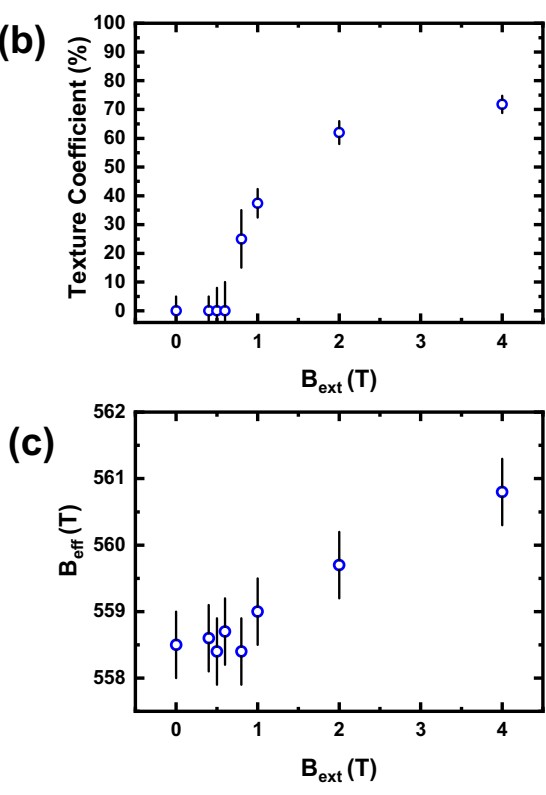

**Fig. 3 | $^{161}$Dy Synchrotron Mössbauer Spectroscopic Data for 1. a** Time-domain SMS spectra taken at $T = 4.2$ K under a range of external magnetic fields applied parallel to the synchrotron radiation. The error bars reflect the standard deviation sigma of the counts N according to radiative processes, which is the square root of N, because of the Poisson character of the nuclear absorption processes during the experiment. **b** The field dependence of the texture coefficient from a spectral analysis using CONUSS[22]. The error bars are the +- total error of the texture coefficient determined with CONUSS. **c** Dependency of the hyperfine magnetic field on the external applied field. The error bars of the hyperfine field represent the Full Width at Half Maximum (FWHM) of an assumed Gaussian distribution of the given hyperfine field, which was also obtained with CONUSS.

0.25−0.40, indicating a wide distribution of relaxation times over the temperature range 1.8–6 K.

The temperature dependence of the relaxation time was fitted using Eq. (1):

$$\tau^{-1} = V * \frac{\exp\left(\frac{\omega}{kT}\right)}{\left(\exp\left(\frac{\omega}{kT}\right) - 1\right)^2} + \tau_0^{-1} * \exp\left(\frac{U_{eff}}{kT}\right) \qquad (1)$$

Taking both Raman and Orbach processes into account, the best fit was obtained using the following parameters: $V = 586$ s$^{-1}$, $\omega = 3.7$ K (2.6 cm$^{-1}$), $\tau_0 = 5.2 \times 10^{-6}$ s and $U_{eff} = 28.8$ K (20.0 cm$^{-1}$) where $\omega$ corresponds well to the exchange KDs calculated using ab initio calculations (see Supplementary Table S18) and where $U_{eff}$ is in line with the energies of the first excited Dy$^{III}$ KDs (see Supplementary Table S6).

**Time-domain Synchrotron Mössbauer Spectroscopy (SMS)**
The $^{161}$Dy time-domain SMS experiments were carried out at the Advanced Photon Source (APS) at Argonne National Laboratory at the beamline 3-ID-D (High-Resolution X-Ray Scattering). Using a He-bath cryostat equipped with a split-coil superconducting magnet (Spectromag 4000 from Oxford Instruments), time-domain SMS spectra were recorded at $T = 4.2$ K with a powder sample of **1** immersed in liquid helium. The external magnetic field oriented parallel to the synchrotron radiation direction was varied up to $\mathbf{B}_{ext} = 4$ T.

The magnetic field dependent $^{161}$Dy time-domain SMS spectra of **1** are displayed in Fig. 3a. Each spectrum shows an exponential decay modulated by a regular beating structure with a beating period of about 5 ns. Time-domain SMS records the time-dependence of the delayed forward scattered quanta after excitation by the synchrotron radiation. The characteristic beating structure is caused by the coherent superposition of quanta with various frequencies due to the hyperfine splitting of the $^{161}$Dy nuclear transition[19–21]. In the case presented here, the beating results solely from the magnetic hyperfine interactions, as previously described in our pilot study presenting the $^{161}$Dy time domain SMS as a spectroscopic tool for the investigation of SMMs[12].

The spectral analysis of the time-domain SMS spectra presented here was achieved by theoretical simulations of the coherently scattered radiation involving the diagonalization of the nuclear $^{161}$Dy Hamiltonian as implemented in the software package CONUSS[22]. The best reproduction of the experimental spectra is based on taking only one $^{161}$Dy site with a static hyperfine interaction and a small Gaussian distribution into account[12]. The approach of only one Dy$^{III}$ site reflects the crystallographic $C_3$ site symmetry of **1**, resulting in one internal hyperfine magnetic field being the same for all of the three Dy$^{III}$ sites.

The 4.2 K time-domain SMS spectrum taken without an external field can be reproduced with an internal hyperfine magnetic field of $\mathbf{B}_0 = 558.5\ (\pm 0.5)$ T with $\sigma(\mathbf{B}_0) = 2.0\ (\pm 0.2)$ T, being close to reported values of 559.8 T[23,24] or 565 T[25] for the free ion Dy$^{III}$ with a $^6H_{15/2}$ ground

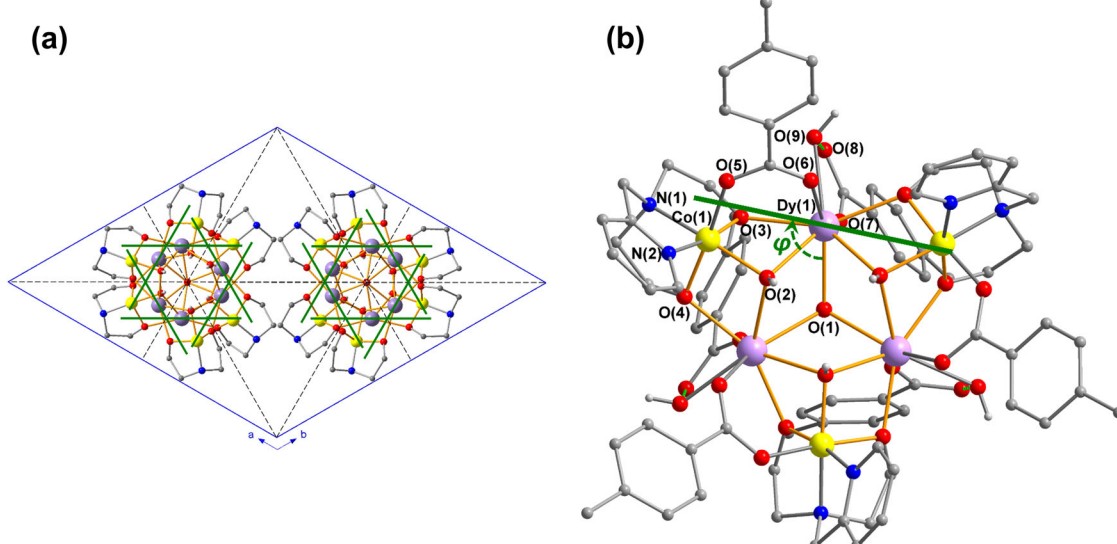

**Fig. 4 | Orientations of the Dy^III Easy Axes in 1. a** Orientations of the cores of the four clusters in the unit cell viewed down the *c*-axis; the crystal *c*-glide planes (here seen edge on) are shown as black dashed lines. **b** Definition of $\varphi$ as the angle by which the projection of a Dy^III easy axis onto the Dy$_3$ plane is rotated away from the vector joining that Dy^III and the centroid of the Dy$_3$ triangle. Dy: violet, Cl: green, O: red, H: white, N: blue, C: gray; Dy^III easy axes shown as green bars.

state (see Supplementary Fig. S9). This free ion Dy^III hyperfine field originates mainly from a large positive contribution due to the partly filled 4 f electron shell with a small negative core polarization field (Fermi-contact term)[26,27]. The obtained result for the hyperfine field is in contrast to the recently reported results on a Dy^III-based single-ion magnet with a hyperfine field exceeding the free ion value[12], attributed to a reduced Fermi-contact term due to the influence of the coordinating ligands. However, for the present polynuclear compound, an additional contribution to the total resulting magnetic hyperfine field may arise due to intramolecular Dy-Dy interactions as the Dy- Dy distance is only 3.85 Å. Therefore, the influence of the coordinating ligands may be less pronounced for **1** or even canceled by intramolecular magnetic interactions.

On application of an external field **B**$_{ext}$, the time evolution of the beating structure hardly changes and is independent of the strength of the magnetic field (see Fig. 3a). However, starting from **B**$_{ext}$ = 0.8 T the amplitude of the beating slowly decreases, especially observable in the range of 30 to 60 ns (Fig. 3a). A rather similar magnetic-field dependent behavior was also observed for the mononuclear single-ion magnet incorporating Dy^III [12]. The spectra taken under the application of an external field were also analyzed assuming one Dy^III site with a static hyperfine interaction and a small Gaussian distribution. The analysis is based on the assumption of an effective magnetic hyperfine field, resulting from a vectorial superposition of the internal and external magnetic field[12]. Due to the well-defined polarization of the synchrotron radiation, the orientation of this effective field influences the probability of the nuclear transitions taking place during the scattering process[21]. In order to consider the influence of the increasing macroscopic magnetization under the application of an external field, a magnetic texture coefficient reflecting the ratio of uniaxial to randomly oriented hyperfine fields was introduced (Fig. 3b and Supplementary Table S4). The magnetic texture coefficient models the sample's magnetization from the viewpoint of the ^161Dy nuclei. This simulation model was reported in detail in our study on the mononuclear SMM[12].

The analysis of the spectra measured under a small external magnetic field **B**$_{ext}$ < 0.8 T reveals, within experimental error, the same parameters as the ones obtained from the spectra measured without a field. The decrease of the amplitude of the beating structure that occurs clearly upon application of a field of **B**$_{ext}$ = 0.8 T can be

reproduced with a non-zero (25 (±10) %) magnetic texture (Supplementary Table S4). A further increase of **B**$_{ext}$ leads to a larger magnetic texture coefficient of 72 (±3) % at 4 T (Supplementary Table S4). Although the powder sample was closely packed in the sample holder, we cannot totally exclude the alignment of microcrystallites in these field-dependent ^161Dy time-domain SMS experiments; the results are in excellent agreement with the magnetization measurements (see above).

This behavior implies a random distribution of the magnetic hyperfine fields at zero-field originating from the random orientation of all microcrystallites in the investigated powder sample. The observed critical field of **B**$_{ext}$ > 0.6 (±0.2) T needed to increase the magnetic texture matches the inflection point in the magnetization curves. Thus, the unusual step-wise behavior of the magnetic hyperfine field texture indicates that there is a critical field (here 0.6 T) required to increase the texture from zero. This behavior is reminiscent of what is observed in the magnetization data for toroidal systems and is also consistent with the magnetization data measured using micro-SQUID presented below. Remarkably, the crossover between the two regimes appears very abrupt in the hyperfine field despite the data being collected at a relatively high temperature (4.2 K).

Regarding the magnitude of the effective magnetic field, an increase of **B**$_{eff}$ with increasing **B**$_{ext}$ is observed (see Fig. 3c), reaching 560.8 (±0.5) T at **B**$_{ext}$ = 4 T, originating from the entirely positive hyperfine field at the ^161Dy nucleus, which means that the Dy magnetic moment and hyperfine field are aligned parallel. Thus, the hyperfine field texture and the increasing magnetization are directly connected[28].

## Measurements on oriented single crystals

Single-crystal magnetic measurements provide direct orientational information of the magnetic anisotropy at the molecular level. The compound described here crystallizes in the trigonal space group $P\bar{3}c1$ with the cluster molecule and one of the chloride counterions on special positions with $C_3$ site symmetry, which simplifies the interpretation of the results. However, a potential complication is introduced by the second, more distant and three-fold disordered chloride. This has implications for the interpretation of the results of single-crystal studies and theoretical calculations, since for any given cluster molecule, this second chloride must occupy one of the three possible

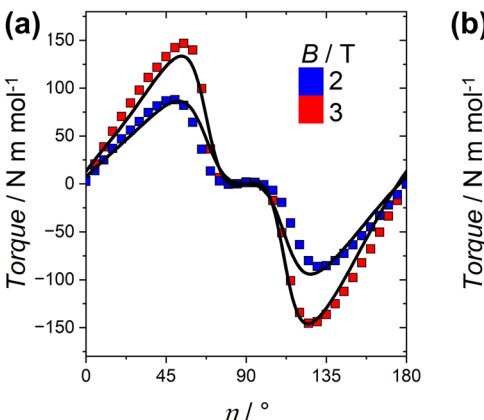
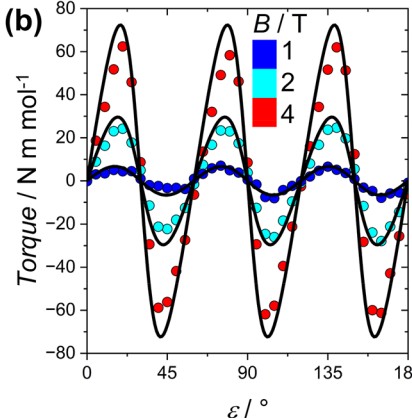

**Fig. 5 | Cantilever Torque Data for 1.** Torque signal recorded at $T = 1.9$ K and various fields (**a**) during the rotation from plane to axis (Rot 1, squares) and (**b**) during the in-plane rotation (Rot 2, circles). The lines are the best fits to the experimental data (see text).

sites. This leads to a breaking of the local threefold symmetry of the cluster. Whether this symmetry-breaking is significant in terms of the magnetic properties of the cluster is investigated and discussed further in the section describing the computational studies on the system.

The relative orientations of the four clusters in the unit cell are dictated by the inversion centers and $c$-glide planes of the structure, as shown in Fig. 4a. The molecules form stacks along the threefold axes, with the molecules in each stack related by the $c$-glide planes. Within a unit cell, two molecules belonging to one such stack are related to the other two in the same cell by the inversion center located at the center of the cell. Thus, in the simulations used here, it is necessary to sum over the anisotropy axes of the six $Dy^{III}$ belonging to two molecules related by the glide planes, since the direction of the easy axis of each $Dy^{III}$ is not restricted by crystal symmetry. The summation over two such molecules and six $Dy^{III}$ ions suffices due to the presence of an inversion center that makes the other two molecules in the unit cell magnetically equivalent.

It is convenient to define the orientation of a $Dy^{III}$ easy axis relative to the molecular geometry in terms of two angles[9]: $\varphi$ being the angle by which the projection of a $Dy^{III}$ easy axis on to the $Dy_3$ plane is rotated away from the vector joining that $Dy^{III}$ ion and the centroid of the $Dy_3$ triangle (Fig. 4b), and $\theta$ being the angle by which the axis is tipped out of the $Dy_3$ plane.

### Cantilever torque magnetometry

The anisotropy of compound **1** has been unraveled using cantilever torque magnetometry. This technique has been used to experimentally detect the anisotropic part of the free energy of both mononuclear and polynuclear lanthanide complexes[29]. In the case of polynuclear complexes, this technique can give access to the anisotropy of the single ions, when sufficiently high fields are applied[30,31]. This feature makes it complementary to micro-SQUID measurements.

Two rotations were performed. In the first rotation (Rot1, quantified by the angle $\eta$), the sample was rotated clockwise out of the $ab$ plane towards the $c$ axis. For the second rotation (Rot2, quantified by angle $\varepsilon$) the sample was rotated about the c axis (i.e., the ab plane was scanned). See Supplementary Table S5 and Supplementary Fig. S10 for further information. The phase of the signal in Rot1 can be easily related to an overall "easy plane" anisotropy of the complex. In other words, the easy axis of each individual $Dy^{III}$ ion must be closer to the $ab$ plane than to the $c$ axis. The plateau at ca. 90° is the characteristic signature of the noncollinearity of the easy axes of the $Dy^{III}$ ions (Fig. 5a). In contrast, the threefold oscillations of Rot2 (Fig. 5b) are the signature of the collinearity between the anisotropy axes of the two glide plane related triangles in the unit cell[30,32]. Data measured under other fields and temperatures are shown in Supplementary Fig. S11.

Each $Dy^{III}$ ion was modeled with an effective $S = 1/2$ and an axial $g$ tensor, and the torque data were satisfactorily simulated using the following spin Hamiltonian (Eq. (2)), where the summation runs over the six equivalent centers belonging to two triangles related by the glide plane:

$$\widehat{H} = \mu_B \sum_{i=1}^{6} \{g_{\parallel} \cdot H_z \cdot S_{zi} + g_{\perp}(H_x \cdot S_{xi} + H_y \cdot S_{yi})\} \quad (2)$$

The simulation provides the orientation of the single ion easy axes. The best simulation (black curves in Fig. 5) was obtained with $g_{\perp} = 2.5$ and $g_{\parallel} = 17$ and gave an angle of 71° between the $c$ axis and the $z$ magnetic axis and an angle of 27° between the crystal $a$ axis and the projection of the $Dy^{III}$ $z$ magnetic axis in the $ab$ plane. These angles correspond to $\theta = 19°$ and $\varphi = 87°$ as defined in Fig. 4b. It is important to note that for a system with more than one inequivalent magnetic center, the number of solutions that torque evaluation provides is equal to the number of inequivalent centers. In this case, the $Dy^{III}$ $z$ axes are almost co-parallel to the glide planes, making the two triangles essentially magnetically equivalent and reducing *de facto* the number of solutions to three. One of the three solutions is compatible with a toroidal arrangement of the magnetic moments, as shown in Fig. 6a, b. However, one must note that the above ambiguity can only be resolved in combination with the theoretical calculations (see below)[33,34].

### Micro-SQUID measurements

We explored the magnetic behavior of **1** further by using a micro-SQUID array on an oriented crystal with the field applied along the $ab$-plane at different temperatures and sweep rates of the applied magnetic field.

The curves exhibit double S-shaped loops, which are symmetric with respect to positive and negative fields. These are attributed to exchange biasing of antiferromagnetically coupled $Dy_3$ in **1** (Fig. 7 and Supplementary Fig. S12).

A minor QTM event connected to a small hysteresis can be attributed to the small components of the three Dy magnetic moments perpendicular to the $ab$-plane, with these components being co-parallel with the molecular threefold axis. The field sweep rate-independence of the step positions, and the step transitions become less pronounced at higher temperatures, indicating QTM as the origin for all three steps. The mean exchange field ($H_{ex}$) can be determined from the inflection points occurring at about $\pm 0.5$ T (Fig. 7c), yielding the coupling constant $J = -0.002$ cm$^{-1}$ using $H_{ex} = 2 \cdot J \cdot m_J / g_J \mu_B$, where $m_J = 15/2$ and $g_J = 4/3$. These results are in line with those of the dc susceptibility studies.

Angular-dependent micro-SQUID measurements, that is the change of magnetization in the $ab$-plane for different directions with

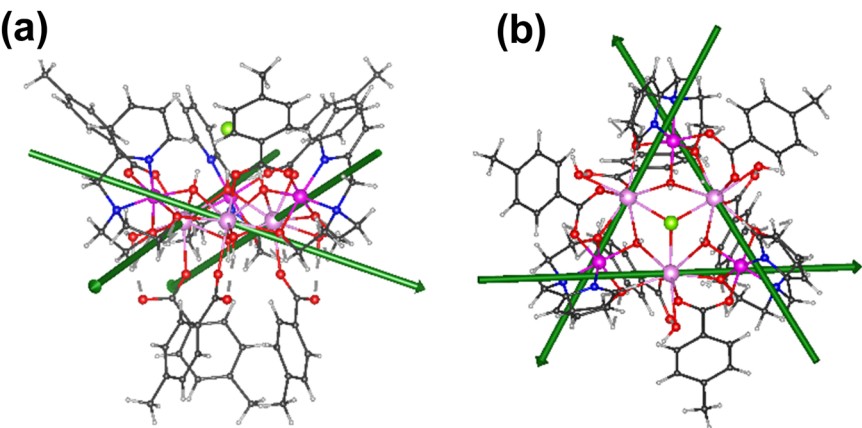

**Fig. 6 | Set of Orientations of the Dy$^{III}$ Easy Axes in 1 consistent with Torque Measurements. a** One of the possible solutions of the torque measurements compatible with a toroidal arrangement of the magnetic moments, (**b**) an alternative view of (**a**) along the crystal *c* axis.

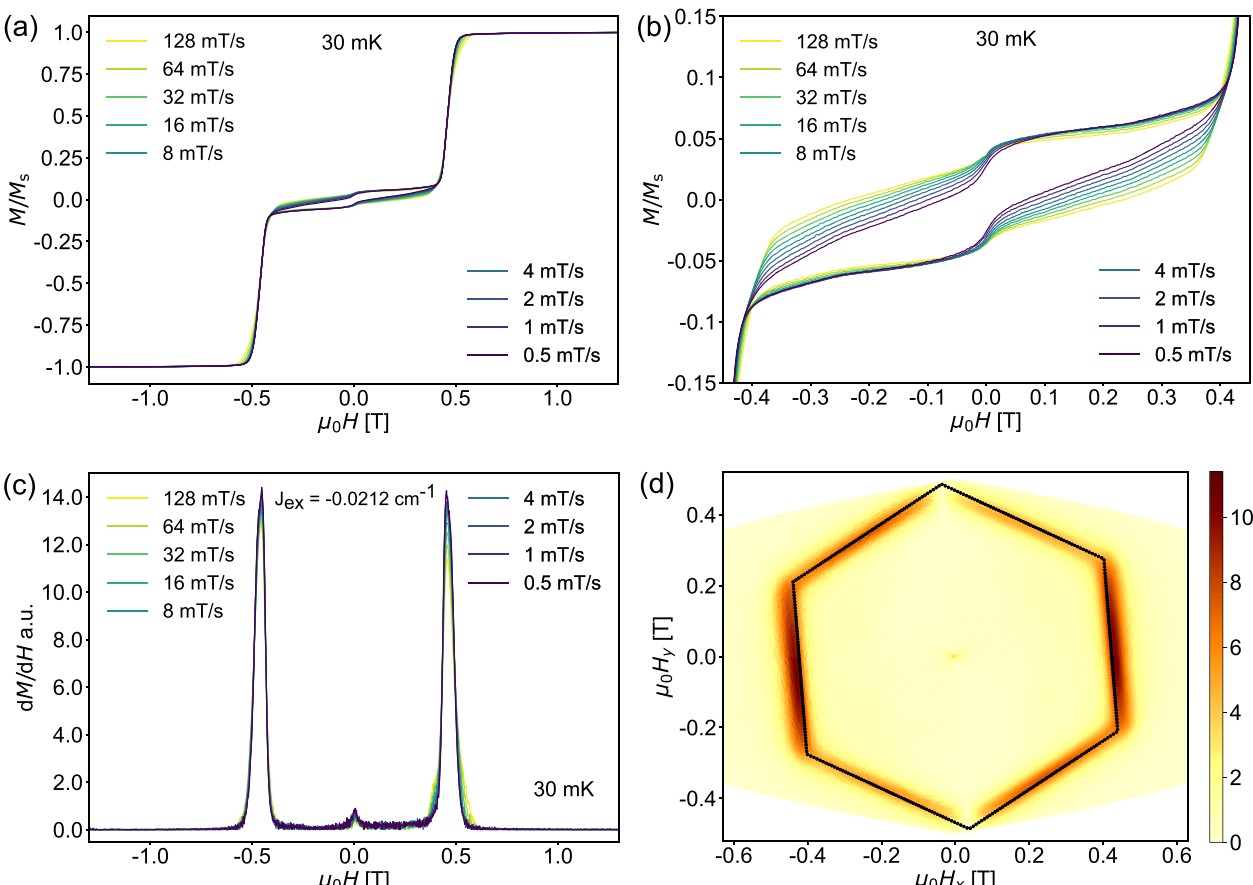

**Fig. 7 | Micro-SQUID Data for 1. a** Field dependence of the magnetization at $T = 0.03$ K with the field applied parallel to the *ab*-plane of the crystal. **b** Hysteresis loops at $T = 0.03$ K between ±0.6 T showing the staircase-like structure. **c** First derivative of the magnetization of the micro-SQUID loops with different scan rates at 0.03 K. **d** The perfect hexagon resulting from micro-SQUID measurements.

respect to the crystal, reveal a hexagonal symmetry, which reflects the $C_3$-symmetry of compound **1**. Figure 7d depicts the derivative of the magnetization intensity, $\delta M/\delta H$, (along the z-axis) upon different field orientations ($\mu_0 H_x/\mu_0 H_y$). Such a perfect hexagon might be expected for a simpler system with a single Dy$_3$ triangle. However, the additional glide planes in the structure of **1** would be expected to double the number of independent directions of Dy$^{III}$ easy axes in the unit cell. The observed "perfect hexagon" is therefore experimental evidence that the anisotropy axes are in fact orientated so that they are closely parallel to the *c*-glide planes in the crystal. As was the case for the torque measurements, for a given Dy$^{III}$ this requirement gives a series of possible axis orientations with their projections in the *ab*-plane rotated by 60° from each other. One of these solutions corresponding to $\varphi = 81°$ is consistent with both the cantilever torque measurements and the ab initio results.

Further measurements, in which the Dy$_3$ plane was mounted vertically, and the out-of-plane to in-plane angular dependence of $H_{sw}$ was determined, gave a value of $\theta = 11°$.

## Computational results

To investigate the magnetic properties of **1** computationally, CASSCF/SO-RASSI calculations were carried out for three different model systems, namely **[Dy$_3$Co$_3$]$^{2+}$**, **[Dy$_3$Co$_3$Cl]$^+$**, and **[Dy$_3$Co$_3$Cl]Cl** (see supplementary information). **[Dy$_3$Co$_3$]$^{2+}$** represents the cationic cluster without the inclusion of any Cl$^-$ counter anions, whereas **[Dy$_3$Co$_3$Cl]$^+$** and **[Dy$_3$Co$_3$Cl]Cl**, include one and two Cl$^-$ anions, respectively (Supplementary Fig. S12). By investigating the three different model systems with charges 2 + , 1 + and 0, respectively, we were able to study the influence of counter Cl$^-$ anions on the magnetic properties of **1**. It can be noted here that the model systems, which include Cl$^-$ anion(s), **[Dy$_3$Co$_3$Cl]$^+$** and **[Dy$_3$Co$_3$Cl]Cl**, are more representative of the species in the crystal structure than **[Dy$_3$Co$_3$]$^{2+}$**. Although **[Dy$_3$Co$_3$]$^{2+}$** and **[Dy$_3$Co$_3$Cl]$^+$** are consistent with the threefold symmetry axis passing through the central Cl$^-$ counter anion and the bridging $\mu_3$-OH unit, the second peripheral Cl$^-$ anion in **[Dy$_3$Co$_3$Cl]Cl** breaks the local threefold symmetry for any given molecule. Thus, the individual CASSCF/SO-RASSI calculations were performed for all three Dy$^{III}$ centers of each of the three model systems without any symmetry idealization.

The calculated energy spectra of the lowest eight Kramers' doublets (KDs) arising from the ground $^6H_{15/2}$ multiplet of each individual Dy$^{III}$ ion in **[Dy$_3$Co$_3$]$^{2+}$**, **[Dy$_3$Co$_3$Cl]$^+$**, and **[Dy$_3$Co$_3$Cl]Cl** are listed in Supplementary Table S6. It is evident that in all three model systems, the total splitting of the ground $^6H_{15/2}$ multiplet is relatively small (< 330 cm$^{-1}$) compared to the systems in which Dy$^{III}$ ion are in a highly axial coordination environment[2,3,35–38]. The stabilization of the ground state is also small in all three studied model systems, with the first excited KD (KD2) lying approximately 15 cm$^{-1}$, 30 cm$^{-1}$, and 26–50 cm$^{-1}$ higher in energy than the ground KD (KD1) in **[Dy$_3$Co$_3$]$^{2+}$**, **[Dy$_3$Co$_3$Cl]$^+$**, and **[Dy$_3$Co$_3$Cl]Cl**, respectively. Similar trends have been observed in the energy spectra of other heterometallic 3d-4f clusters[31,39]. Furthermore, for **[Dy$_3$Co$_3$]$^{2+}$** and **[Dy$_3$Co$_3$Cl]$^+$** the calculated energy spectrum of each individual Dy$^{III}$ ion clearly indicates that the coordination environment around each of the three Dy$^{III}$ ions is identical in **[Dy$_3$Co$_3$]$^{2+}$** and **[Dy$_3$Co$_3$Cl]$^+$** as expected due to their threefold symmetry. When the energy spectra of **[Dy$_3$Co$_3$]$^{2+}$**, **[Dy$_3$Co$_3$Cl]$^+$**, and **[Dy$_3$Co$_3$Cl]Cl** are compared, it can be seen that both Cl$^-$ anions affect the splitting of energy levels. The Cl$^-$ anion that is 4.526 Å away from three Dy$^{III}$ ions on the C$_3$ axis of **[Dy$_3$Co$_3$Cl]$^+$** stabilizes the ground state by ~ 15 cm$^{-1}$ and slightly lowers the energies of excited KDs except the energy of the seventh excited KD (KD8). In contrast, the peripheral Cl$^-$ anion in **[Dy$_3$Co$_3$Cl]Cl**, with the nearest Cl-Dy$^{III}$ distance of 8.77(2) Å, breaks the three-fold symmetry, and the energies of the corresponding KDs of the individual Dy$^{III}$ ions vary from 2 to 22 cm$^{-1}$ in **[Dy$_3$Co$_3$Cl]Cl**. This indicates that the influence of the relatively remote Cl$^-$ counter anion on the splitting of energy levels of each individual Dy$^{III}$ center is small but not negligible.

In the three model systems, the principal magnetic axis of the ground KD of each Dy$^{III}$ ion is oriented with the out-of-plane angle $\theta$ at ~ 13° for **[Dy$_3$Co$_3$Cl]$^+$** and for **[Dy$_3$Co$_3$Cl]Cl** and ~ 18° for **[Dy$_3$Co$_3$]$^{2+}$** with respect to the plane of the Dy$_3$ triangle (Supplementary Fig. S13 and Supplementary Table S7). In particular, the calculated angles for **[Dy$_3$Co$_3$Cl]$^+$** and **[Dy$_3$Co$_3$Cl]Cl** are in good agreement with the out-of-plane angle $\theta$ determined from the micro-SQUID measurements (~11°) and not too far from that obtained from torque magnetometry (~ 19°). For the orientations of the easy axes in the *ab* plane, the calculations for **[Dy$_3$Co$_3$Cl]$^+$** and **[Dy$_3$Co$_3$Cl]Cl** are close to those obtained using torque magnetometry and micro-SQUID (compare Figs. 4a, 6 and Supplementary Fig. S14). A full comparison of calculated and experimental axes are given in Table 1.

**Table 1 | Comparison of the orientations of the easy axes of the ground Kramers' doublets of the Dy$^{III}$ ions in 1 relative to the molecular framework obtained from the three theoretical models considered here, with those from the angle-dependence of the microSQUID and cantilever torque measurements**

| | $\varphi$ / ° | $\theta$ / ° |
|---|---|---|
| **[Dy$_3$Co$_3$]$^{2+}$** | 66 | 17.8 |
| **[Dy$_3$Co$_3$Cl]$^+$** | 86 | 12.7 |
| **[Dy$_3$Co$_3$Cl]Cl** | 86, 87, 86 | 12.8, 13.7, 12.9 |
| microSQUID | 81 (± 60n) | 11 |
| torque magnetometry | 87 (± 60n) | 19 |

The angles $\varphi$ and $\theta$ are as defined in the text.

Moreover, it is clear from Supplementary Fig. S14 that the main magnetic axes of the three Dy$^{III}$ ions in all three model systems adopt the typical orientation of the magnetic moments observed for systems with an axial toroidal moment, oriented in a propeller-like arrangement[11,16,40–42]. The calculated g tensors of the individual Dy$^{III}$ ions (see Supplementary Table S8) indicate that the ground KDs of the Dy$^{III}$ ions in **[Dy$_3$Co$_3$Cl]$^+$** and **[Dy$_3$Co$_3$Cl]Cl** have small transverse components ($g_x = 0.3$, $g_y = 0.5$, $g_z = 18.5$ for **[Dy$_3$Co$_3$Cl]$^+$** and $g_x = 0.08$–$0.38$, $g_y = 0.13$–$0.78$, $g_z = 18.28$–$18.97$ for **[Dy$_3$Co$_3$Cl]Cl**). Slightly different values were obtained from the torque simulation ($g_x = g_y = 2.5$, $g_z = 17$). However, the exclusion of both Cl$^-$ anions from the calculations significantly decreases the axiality of all three Dy$^{III}$ ions as illustrated by the calculated g tensors of the ground KD of **[Dy$_3$Co$_3$]$^{2+}$** ($g_x = 0.8$, $g_y = 3.6$, and $g_z = 15.5$)[43–46]. Based on the calculated transverse components of the ground KDs of each Dy$^{3+}$ ion, it can be concluded that they are sufficiently large to promote quantum tunneling of the magnetization in **1** in the absence of an external magnetic field, consistent with the experimental data.

The quadratic decomposition of the SO-RASSI wave functions of the eight lowest KDs projected on to the |$J,M_J$> states ($J = 15/2$) allows us to determine the mixing of states in the ground $^6H_{15/2}$ multiplet of the Dy$^{III}$ ions (Supplementary Tables S10–S18). For **[Dy$_3$Co$_3$Cl]Cl**, the calculations reveal that the ground KDs of Dy1, Dy2, and Dy3 contain projections mainly on to the $M_J = \pm 15/2$ state with the following values 0.82, 0.82, and 0.71, respectively, as well as smaller projections from the several excited states (see Supplementary Tables S10–S12). In the case of **[Dy$_3$Co$_3$]$^{2+}$** and **[Dy$_3$Co$_3$Cl]$^+$**, the decomposition of the ground KDs of each Dy$^{III}$ ion is also a mixture of $M_J = \pm 15/2$ states and excited states with varying contributions (see Supplementary Tables S13–S18). In all these model systems, the excited KDs are strongly mixed and cannot be described in terms of any pure $M_J$ state (see Supplementary Tables S10–S18). As a whole, the calculated g tensors, their orientations, and the quadratic decomposition of the SO-RASSI wave functions for **[Dy$_3$Co$_3$]$^{2+}$**, **[Dy$_3$Co$_3$Cl]$^+$**, and **[Dy$_3$Co$_3$Cl]Cl** are consistent with low symmetry coordination environments for all Dy$^{III}$ ions in **1**.

The total magnetic interactions between the Dy$^{III}$ ions in **1** were evaluated by calculating the exchange and dipolar interactions in **[Dy$_3$Co$_3$Cl]$^+$** and **[Dy$_3$Co$_3$Cl]Cl** using the non-collinear Ising model (see Supporting Information). Such a model is valid when the investigated system is strongly axial. The latter is true for [Dy$_3$Co$_3$Cl]$^+$ and [Dy$_3$Co$_3$Cl]Cl but not for [Dy$_3$Co$_3$]$^{2+}$, which strongly deviates from the perfect axial system with $g_x = g_y = 0$, and $g_z = 20$. For this reason, the total magnetic interactions were not evaluated for **[Dy$_3$Co$_3$]$^{2+}$**.

To obtain values for $J_{Ising\ exch}$ using Supplementary Equation S2, we first evaluated the exchange interaction between Dy$^{III}$ ions using the Lines model with an effective Heisenberg Hamiltonian (Supplementary Equation S1)[47]. The Lines exchange parameters were obtained by fitting the calculated susceptibility and magnetization

data to the experimental values. To restrict the number of fitting parameters to one, we assumed that the exchange interaction is identical for all three interacting pairs. This assumption holds for the symmetric **[Dy$_3$Co$_3$Cl]$^+$** but introduces some minor errors for **[Dy$_3$Co$_3$Cl]Cl**. As shown in Supplementary Fig. S14, using the Lines exchange parameters $-8.9 \times 10^{-2}$ cm$^{-1}$ and $-9.4 \times 10^{-2}$ cm$^{-1}$ for **[Dy$_3$Co$_3$Cl]$^+$** and **[Dy$_3$Co$_3$Cl]Cl**, respectively, a very good fit between the experimental and computational data is obtained. The negative Lines exchange parameters, that highlight the antiferromagnetic arrangement of the spins, were converted to $J_{Ising\_exch}$ using Supplementary Equation S2 that takes into account the angle δ between the main magnetic axis of the interacting sites[16]. Since for all interacting pairs δ is ~115°, the relation (Supplementary Equation S2) between the Lines and Ising exchange parameters becomes $J_{Ising\_exch} \approx$ -10.75 $J_{Lines\_exch}$. Thus, $J_{Ising\_exch} > 0$ for the antiferromagnetic $J_{Lines\_exch}$, i.e., we note that the sign convention of the Lines model differs from the Ising one in this particular case. By employing Supplementary Equation S2 for all interacting pairs $J_{Ising\_exch}$ was found to be ~1 cm$^{-1}$ (Table 2). The exchange interaction is roughly one third of the dipolar interaction that was calculated to be ~3 cm$^{-1}$ between all interacting pairs using Supplementary Equation S4 (Table 2). Thus, the total interaction between Dy-Dy pairs is ~4 cm$^{-1}$.

To investigate the influence of the applied magnetic field on the exchange interaction, we calculated the evolution of four exchange KDs of **[Dy$_3$Co$_3$Cl]$^+$** with an applied field (Fig. 8). These four exchange doublets arise from the interaction between the ground KDs of three Dy$^{III}$ ions. Without an applied external field, the three lowest excited

exchange KDs are 3.68 cm$^{-1}$, 3.70 cm$^{-1}$, and 3.71 cm$^{-1}$ higher in energy than the ground exchange KD (Fig. 8 and Supplementary Table S19). The result is in good agreement with the calculated $J_{Ising\_tot} \approx 4.0$ cm$^{-1}$ since the total coupling constant should correlate with the excitation energies of the lowest excited exchange KD. Interestingly, the calculated g-tensor for the ground exchange doublet shows a magnetic ground state for **[Dy$_3$Co$_3$Cl]$^+$** of $g_x = g_y = 0$ and $g_z = 12.22$ with the magnetic axis oriented perpendicularly to the triangular Dy$_3$ plane[11,16,40,41]. The calculated result is in line with the experimentally and computationally determined $\chi T$ curves that do not go to zero at low temperature. Due to the orientation of the main exchange magnetic axis, the ground exchange KD splits essentially linearly when the external magnetic field is applied perpendicular to the triangular Dy$_3$ plane, whereas applying the field along the triangular Dy$_3$ plane hardly affects the splitting of the ground exchange KD. Importantly, when the field is applied in the triangular Dy$_3$ plane, the crossing of levels occurs around 0.5 T, in close agreement with the inflection point determined from the SMS and magnetization measurements.

[Co$^{III}$$_3$Dy$^{III}$$_3$(μ$_3$-OH)$_4$(O$_2$C-C$_6$H$_4$-$p$-Me)$_6$(pmide)$_3$(H$_2$O)$_3$]Cl$_2$·10MeCN **(1)** with its shuttlecock structure is an aesthetically pleasing new addition to the family of heterometallic 3d-4f Single Molecule Toroics (SMTs)[42,48,49]. We report that $^{161}$Dy-SMS measurements on powder samples obtained under applied fields show the transition from the essentially non-magnetic ground state to an excited magnetic state. These are in line with the results of the SQUID data, but show a more abrupt transition than that observed in the SQUID magnetization measurements at the same temperature, which is advantageous for the accurate determination of the position of the level crossing. This is due to the short experimental time window of $^{161}$Dy-SMS (~1 ns), which is given by the Larmor precession time of the $^{161}$Dy nucleus in its hyperfine field.

The magnitude of the hyperfine field is directly related to the magnetic moment of in this case the Dy$^{III}$ ion, however, its magnitude is independent of the Boltzmann population of the $m_J$ states. Nevertheless, using the texture parameter, a Boltzmann population of the $m_J$ states can be modeled, which also gives access to the magnetization of the sample in the ns time window.

Single crystal experiments using torque cantilever and micro-SQUID magnetometry were used to unravel the orientation of the anisotropy axes of the Dy$^{III}$ ions, revealing a toroidal spin structure. The results from these two techniques were in good agreement with each other and with theoretical calculations. These calculations showed the importance of including the strongly hydrogen-bonded chloride counter anion in the model, whereas the inclusion of the second and more remote chloride counter anion was explored and found to have a small but not insignificant impact on the results.

**Table 2 | Calculated Lines ($J_{Lines\_exch}$) and Ising ($J_{Ising\_exch}$, $J_{Ising\_dip}$, and $J_{Ising\_tot}$) parameters between Dy1–Dy2, Dy1–Dy3, and Dy2–Dy3 ions within [Dy$_3$Co$_3$Cl]Cl and [Dy$_3$Co$_3$Cl]$^+$, as well as the respective calculated angle δ between the main magnetic axes of two interacting sites**

| [Dy$_3$Co$_3$Cl]Cl | $J_{Lines\_exch}$ (cm$^{-1}$) | δ (°) | $J_{Ising\_exch}$ (cm$^{-1}$) | $J_{Ising\_dip}$ (cm$^{-1}$) | $J_{Ising\_tot}$ (cm$^{-1}$) |
|---|---|---|---|---|---|
| Dy1–Dy2 | − 0.094 | 114.399 | 0.971 | 2.920 | 3.891 |
| Dy1–Dy3 | − 0.094 | 115.010 | 0.994 | 3.000 | 3.994 |
| Dy2–Dy3 | −0.094 | 115.630 | 1.017 | 2.919 | 3.935 |
| [Dy$_3$Co$_3$Cl]$^+$ | $J_{Lines\_exch}$ (cm$^{-1}$) | δ (°) | $J_{Ising\_exch}$ (cm$^{-1}$) | $J_{Ising\_dip}$ (cm$^{-1}$) | $J_{Ising\_tot}$ (cm$^{-1}$) |
| Dy1–Dy2 | − 0.087 | 115.332 | 0.931 | 2.901 | 3.832 |
| Dy1–Dy3 | − 0.087 | 115.272 | 0.929 | 2.901 | 3.829 |
| Dy2–Dy3 | − 0.087 | 115.289 | 0.929 | 2.904 | 3.833 |

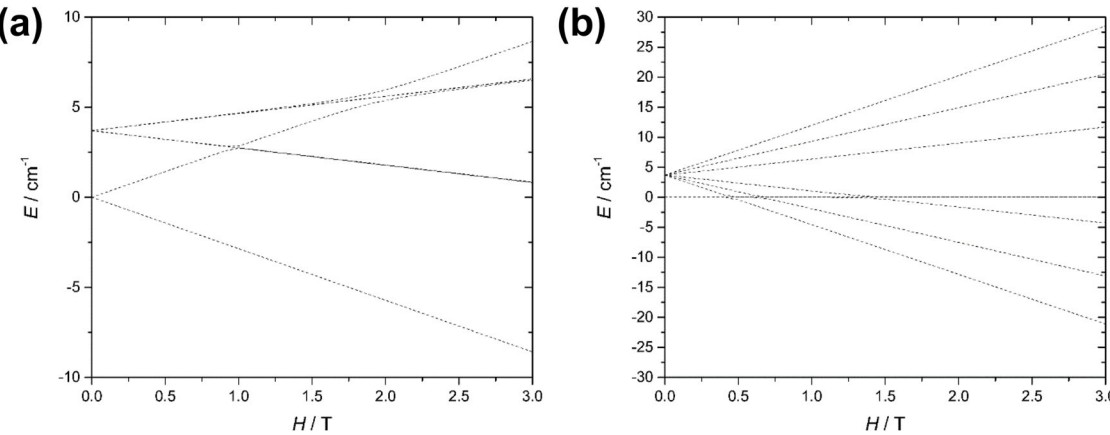

**Fig. 8 | Zeeman diagram for 1.** Evolution of the lowest magnetic states in [Dy$_3$Co$_3$Cl]$^+$ with an applied magnetic field projected to perpendicular to the Dy$_3$ triangular plane (x = 0.0, y = 0.0, z = 1.0; (**a**) and within the Dy$_3$ triangular plane (x = 0.7, y = 0.7, z = 0.0; (**b**).

## Methods

### Synthesis of [Co$^{III}_3$Dy$^{III}_3$(OH)$_4$(O$_2$C-C$_6$H$_4$-$p$-Me)$_6$(pmide)$_3$(H$_2$O)$_3$]Cl$_2$·10MeCN (1)

H$_2$pmide (100 mg, 0.5 mmol) in MeCN (5 ml) was added to a solution of CoCl$_3$·6H$_2$O (66 mg, 0.25 mmol), DyCl$_3$·6H$_2$O (94 mg, 0.25 mmol) and 4-methylbenzoic acid (136 mg, 1 mmol) in a mixture of MeCN (15 ml) and MeOH (5 ml). After stirring for 10 minutes, NEt$_3$ (0.56 ml, 4 mmol) was added, and the solution was stirred for a further 30 minutes. The solution was then filtered and left undisturbed. After three days, purple single crystals in the form of hexagonal plates were collected by filtration. Anal. Calcd for C$_{98}$H$_{124}$Cl$_2$Co$_3$Dy$_3$N$_{16}$O$_{25}$: C 44.23; H 4.70; N 8.42. Found: C 44.14; H 4.57; N 8.52. Selected IR data (KBr, cm$^{-1}$) for **1**: 3493 (br), 3062 (w), 2978 (w), 2856 (m), 1595 (s), 1542 (s), 1097 (s), 908 (s), 719 (s), 673 (s), 593 (s).

**X-ray analysis.** X-ray crystallographic data were collected at 180(2) K on a Stoe IPDS II diffractometer using graphite-monochromated Mo-Ka radiation. The structure was solved by dual-space direct methods (SHELXT)[50] and refined by full-matrix least-squares against F$_o^2$ (SHELXL-2016)[51] within the Olex2 platform[52]. Anisotropic displacement parameters were used for all ordered non-hydrogen atoms. O-H distances were restrained to 0.88(4) Å, and organic hydrogen atoms were placed in calculated positions. One chloride counterion and six lattice MeCN per cluster could be refined anisotropically; the second chloride and four further MeCN were mutually disordered, and these were refined using partial-occupancy isotropic atoms. Full crystallographic data and details of the structural determinations for the structures in this paper have been deposited with the Cambridge Crystallographic Data Center as supplementary publication nos. CCDC 1560928. Copies of the data can be obtained, free of charge, from https://www.ccdc.cam.ac.uk/structures/.

**Magnetic measurements.** The magnetic susceptibility measurements were obtained using a Quantum Design SQUID magnetometer MPMS-XL in the temperature range 1.8–300 K. Measurements were performed on a polycrystalline sample restrained with eicosane in a polyethylene bag. Magnetization isotherms were collected at 2, 3 and 5 K between 0 and 7 T. Alternating current (ac) susceptibility measurements were performed with an oscillating field of 3 Oe and ac frequencies ranging from 1 to 1500 Hz. The magnetic data were corrected for the sample holder and the diamagnetic contribution.

**$^{161}$Dy time-domain SMS.** Time-domain SMS experiments were carried out at the Advanced Photon Source (APS) at Argonne, Illinois, operating in the standard operating mode (top-up, bunch separation of 153 ns). The monochromatization of the desired nuclear resonant transition energy of 25.65 keV was achieved using the suitable monochromator setup as previously described[53,54]. In analogy to our recent experiments[12], the beam size was about 0.5 mm (horizontal) and 0.3 mm (vertical) at the sample position, with a flux of about 1.5 ·10$^8$ phs/s. The polycrystalline compound **1** was pressed into a hole of a copper sample holder with a diameter of 1.5 mm and a thickness of 1.5 mm. The elastic coherent signal was measured in forward geometry with a detector system consisting of Avalanche Photo Diodes and electronic amplification[21,53].

The analysis of the experimental time-domain SMS data was performed by the theoretical calculation of the nuclear forward scattered amplitude implemented in the software CONUSS[22]. The conversion of the raw time-domain SMS data into the time scale is done with a conversion factor of 0.0489 ns per channel and with a binning of three channels. The time zero point is given by the maximum of the prompt signal[21]. This conversion process influences slightly the determination of the magnetic hyperfine field, leading to a maximal error of $\Delta B_{eff} = 0.5$ T.

**Cantilever torque magnetometry.** Torque magnetometry experiments were performed by using a homemade two-legged CuBe cantilever separated by 0.1 mm from a gold plate[29]. The cantilever was inserted into an Oxford Instruments MAGLAB2000 platform with automated rotation of the cantilever chip in a vertical magnet. The capacitance of the cantilever was detected with an Andeen-Hagerling 2500 Ultra Precision Capacitance Bridge.

**Ab initio calculations.** The geometries of model systems were extracted from the crystal structure of **1**, but the methyl groups of $p$-methyl benzoate ligands were replaced with H in all studied complexes to simplify the model systems slightly. Prior to the CASSCF/SO-RASSI calculations, the positions of hydrogen atoms were optimized at the RI-PBEPBE-D3/def2-TZVP level of theory[55–63], using Turbomole V7.3,26 while the positions of other atoms were kept frozen. In this prior optimization step, Dy$^{III}$ ions were substituted for Y$^{III}$ ions to avoid convergence problems, with the core electrons of Y$^{III}$ ions being modeled using an effective core potential[55].

The CASSCF/SO-RASSI calculations were carried out using MOL-CAS 8.4[64], and were performed for each individual Dy$^{III}$ ions separately with the two other Dy$^{III}$ ions replaced with diamagnetic Y$^{III}$ ions. The ANO-RCC-VTZP basis set was used for Dy$^{III}$, while ANO-RCC-VDZP was used for all other atoms (H, C, N, O, and Co) except the C atoms of phenyl rings and all C-H protons for which the ANO-RCC-MB basis set was used[65,66]. To speed up the calculations, the Cholesky decomposition was employed for two electron integrals with the threshold value of 10$^{-8}$. Scalar relativistic effects were taken into account using the exact two-component (X2C) transformation[67,68]. The 9 electrons of a Dy$^{III}$ ion in 7 4f-orbitals give rise to 21 sextets, 224 quartets, and 490 doublets. All these states were solved in the state-average CASSCF calculations. Out of all these calculated states, 21 sextets, 128 spin quartets, and 130 spin doublets were mixed by spin-orbit coupling in the subsequent restricted active space state interaction calculations (SO-RASSI)[69]. From the calculated SO-RASSI wave functions local magnetic properties of each individual Dy$^{III}$ ions were extracted utilizing the SINGLE_ANISO routine[70–72]. To investigate the total magnetic interactions (exchange and dipolar) between the Dy$^{III}$ ions in the model systems [Dy$_3$Co$_3$Cl]$^+$ and [Dy$_3$Co$_3$Cl$_2$], the exchange interaction was first modeled using the Lines model and the dipolar interaction was calculated as implemented in the POLY_ANISO program[16,73].

## Data availability

The crystal structure details have been deposited at the Cambridge Crystallographic Database (CCDC) with the deposition number CCDC 1560928. Copies of the data can be obtained, free of charge, from https://www.ccdc.cam.ac.uk/structures/. The other datasets generated or analyzed during this study are available from the KITopen repository: https://doi.org/10.35097/zwswh43dwg837xgu. All data are available from the corresponding author upon request. Source data are provided in this paper.

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

## Acknowledgements

J.B., C.E.A., R.S. and A.K.P. acknowledge funding by the German Research Council (DFG) via the CRC 1573 "4 f for Future", and are also grateful for funding by the Helmholtz Foundation POF MSE. W.W. thanks the German Research Foundation (DFG) concerning the Gottfried Wilhelm Leibniz-Award, ZVN-2020_WE 4458-5. The support of the Italian MUR through the Dipartimenti di Eccellenza 2023-2027 (DICUS 2.0) project is acknowledged. J.O.M. acknowledges the Research Council of Finland (projects 315829 and 338733) for the financial support, as well as the CSC-IT Center for Science in Finland, the Finnish Grid and Cloud Infrastructure (persistent identifier urn:nbn:fi:research-infras-2016072533) and Prof. Heikki M. Tuononen (University of Jyväskylä) for providing computational resources for the project. L.S. and V.S. thank W. Sturhahn for his support in using the Software CONUSS. V.S. acknowledges funding through DFG CRC/Transregio 173, "Spin + X" and CRC/Transregio 88, "3MET". This research used resources of the Advanced Photon Source, a U.S. Department of Energy (DOE) Office of Science user facility operated for the DOE Office of Science by Argonne National Laboratory under Contract No. DE-AC02-06CH11357.

## Author contributions

Synthesis and characterization: Y.P.; synchrotron time-domain Mössbauer spectroscopy (experiment and interpretation): L.S., H.A., J.A.W., E.E.A., J.Z., W.B., D.E.B. and V.S.; magnetic SQUID measurements (experiment and interpretation): Y.P., J.B., C.E.A., R.S. and A.K.P.; crystallography: C.E.A.; torque magnetometry (experiment and interpretation): L.T., M.P., and R.S.; quantum chemical calculations: J.O.M. and L.F.C.; micro-SQUID measurements (experiment and interpretation): M.S. and W.W.; writing and editing of the manuscript: Y.P., J.B., L.S., C.E.A., J.O.M., M.P., R.S., V.S. and A.K.P.; Funding acquisition and supervision: W.W., R.S., V.S. and A.K.P. All authors have contributed to the preparation of the manuscript.

## Funding

## Competing interests

The authors declare no competing interests.

## Additional information

[1]Institute of Inorganic Chemistry (AOC), Karlsruhe Institute of Technology (KIT), Karlsruhe, Germany. [2]School of Chemistry and Chemical Engineering, Jiangxi Provincial Key Laboratory of Functional Crystalline Materials Chemistry, Jiangxi University of Science and Technology, Ganzhou, Jiangxi Province, P. R. China. [3]Institute of Nanotechnology (INT), Karlsruhe Institute of Technology (KIT), Karlsruhe, Germany. [4]Institute for Quantum Materials and Technologies (IQMT), Karlsruhe Institute of Technology (KIT), Karlsruhe, Germany. [5]Department of Physics, University of Kaiserlautern-Landau, Erwin-Schrödinger-Str. 46, Kaiserslautern, Germany. [6]Department of Radiotherapy and Radiation Oncology, Saarland University Medical Centre, Homburg, Saar, Germany. [7]Institute of Physics (PHI), Karlsruhe Institute of Technology (KIT), Kaiserstr. 12, Karlsruhe, Germany. [8]Advanced Photon Source, Argonne National Laboratory, Argonne, Lemont, Illinois, USA. [9]SmartState Center for Experimental Nanoscale Physics, Department of Physics and Astronomy, University of South Carolina, Columbia, South Carolina, USA. [10]Department of Chemistry "U. Schiff", University of Florence, Sesto Fiorentino, Italy. [11]Institute of Physical Chemistry and Center for Integrated Quantum Science and Technology, University of Stuttgart, Stuttgart, Germany. [12]University of Jyväskylä, Department of Chemistry, Nanoscience Centre, P.O. Box 35, FI-40014 University of Jyväskylä, Jyväskylä, Finland. [13]Department of Physics, Northern Illinois University, DeKalb, Illinois, USA. [14]Theory of Nanomaterials Group, Katholieke Universiteit Leuven, Celesijnenlaan, 200F, Heverlee, Belgium. ✉e-mail: yan.peng@jxust.edu.cn; roberta.sessoli@unifi.it; schuene@rptu.de; annie.powell@kit.edu

