## [Transparent Peer Review file · Nature Communications]

Adding ^{161}Dy -Mössbauer Spectroscopy to a Multitechnique Investigation of Magnetic Transitions in a $\{\text{Co}^{\text{III}}_3\text{Dy}^{\text{III}}_3\}$ Single-Molecule Toroid

Corresponding Author: Professor Annie Powell

Version 0:

Reviewer comments:

Reviewer #1

(Remarks to the Author)

Manuscript # NCOMMS-25-53859 comment

This work by Yan Peng, Roberta Sessoli, Volker Schünemann, Annie K. Powell, and their co-workers describes a hexanuclear lanthanide/cobalt complex exhibiting unique magnetic properties. The authors thoroughly investigated the crystal structures and magnetic properties based on both experimental data and computational results. However, the manuscript would benefit from a more discussion on magneto-structural correlations, which would enhance its value for a broader scientific audience. For this reason, the reviewer would recommend this manuscript for publication in Nature Communications after the following major issues are addressed:

1. The authors used $\text{CoCl}_3 \cdot 6\text{H}_2\text{O}$ as the starting material, and low-spin $\text{Co}(\text{III})$, which is diamagnetic, was identified in the crystal structure. Consequently, the three $\text{Co}(\text{III})$ centers do not contribute to the magnetic properties. Did the authors attempt to use $\text{Co}(\text{II})$ instead of $\text{Co}(\text{III})$, considering that $\text{Co}(\text{II})$ complexes with the same ligand system have demonstrated single-molecule magnet (SMM) behavior?
2. Following the previous comment, if the $\text{Co}(\text{III})$ centers are diamagnetic, this molecule can be considered a Dy_3 triangular complex with threefold site symmetry. The authors should clarify why this particular molecule is unique and significant for the study of magnetic properties.
3. The checkCIF report provided by the authors does not include the FCF file, and the counterions (Cl^-) are not represented with thermal ellipsoids. Could the authors clarify whether there are any issues with the crystal structure? Please provide an explanation.
4. In the ^{161}Dy Mössbauer section, the results appear to differ from those reported for mononuclear systems in previous studies. Can one conclude that the magnetic behavior of this multinuclear system is fundamentally different from that of mononuclear complexes? Were intramolecular interactions, particularly between Dy – Dy centers, critical to the magnetic properties of this multinuclear compound? Moreover, both the SQUID measurements and computational results suggest the presence of antiferromagnetic interactions. In this research, did spin frustration play a significant role in the formation of the toroidal magnetic state observed in this system?
5. The authors performed calculations to compare three analogs: $[\text{Dy}_3\text{Co}_3]^{2+}$, $[\text{Dy}_3\text{Co}_3\text{Cl}]^+$, and $[\text{Dy}_3\text{Co}_3\text{Cl}]\text{Cl}$. Indeed, even minor differences—such as changes in molecular charge or the substitution of counterions—can subtly modify the crystal structure and potentially lead to significant changes in magnetic properties. Did the authors observe any structural differences (e.g., bond lengths or angles) in the optimized geometries prior to performing magnetic calculations? Moreover, the charge and nature of the counterions have also been shown to play an important role in influencing magnetic properties in recent studies. Could the authors comment on how the magnetic behavior might change if the counterion were substituted with a different monovalent anion?

Reviewer #2

(Remarks to the Author)

Powell et al report the structure and properties of the magnetotroic compound abbreviated to Co₃Dy₃. They investigate the magnetotroic (MT) properties using a range of techniques including the infrequently reported synchrotron Moessbauer technique.

Overall, this is a very strong piece of work. There are relatively few MT compounds in the literature and even fewer sit on an ideal threefold rotation site in the crystal lattice (and true threefold symmetry is a requirement). As such this is an ideal example to demonstrate the effectiveness of multiple different techniques including conventional magnetometry, Torque magnetometry and CASSCF calculations.

All techniques are well reported and convincingly demonstrate that this is an MT complex.

The Dy Moessbauer results are a new technique for this field and given the prevalence of Dy in molecular magnets, this is an interesting technique. I particularly appreciated the clarity of the spectra and how obvious the changes to the spectra are with an applied field. The field dependence of the hyperfine field and texture coefficient are clear, and the latter shows a change which is coincident with that observed by magnetometry. I do have a question regarding this field dependence and equivalence with magnetometry below. If I had to have one criticism of this paper and the technique, it is not clear what additional information is learnt from the Dy Moessbauer. What does the change in magnetic texture represent on the atomic level and what information is encoding in this about the magnetoroidal behaviour that aren't accessible with other techniques.

Overall, I think that this is a rare case of an ideal symmetry MT compound that has been thoroughly studied using a number of techniques. The addition of synchrotron Moessbauer is an excellent addition. With my comments below addressed, this should be published in Nature Communications.

I have a small number of scientific and technical points that I would like the authors to address.

1. The primary identification of the presence of MT is the point of inflection in the magnetisation data. The other experimental and theoretical techniques used verify this. This point of inflection is clear in the dM/dH derivative plotted with the 2 K data (figure 2 inset). However, it is not clear that the same point of inflection is present in the higher temperature M/H data and the derivative of those data is not presented. This is important as the synchrotron Moessbauer results are presented at 3.6 K and one of the key observations of the data is that the field dependence is the same as the magnetometry at 2 K. However, as the data is presented it is not clear that true MT behaviour is observed at this temperature – my suspicion is that the higher temperature data will have thermal effects due to population of low lying states, so that the M vs H data will not show a point of inflection. This needs to be addressed to make it clear that this is still an MT at 3.6 K. Or that this additional technique is sensitive to MT at these elevated temperatures.
2. The synchrotron Moessbauer result was done with an unrestrained sample. This is perfectly reasonable as restraining media may give unwanted signals. However, the results are predicated on hoping that the sample was packed tightly enough for reorientation not to occur. I would have liked to see a repeat of the zero-field measurement after the field has been applied to demonstrate the signal was still the same after application of an 4 T field and thus giving some experimental evidence that the sample was stationary.
3. The only verification of bulk sample purity presented is elemental analysis, while these are within the allowed ranges, I would have hoped to see an X-ray powder diffraction pattern to demonstrate bulk purity.

In addition, I have a comment about some of the language used. The term “spin structure” is used multiple times throughout (including in the abstract where it is referred to as a spin structure that can be correlated with the crystal structure). If I understand the properties of these molecules correctly, these are still fluctuating moments and the spin structures are average ones. Some of the language gets very close to that used to describe magnetic structures with long range magnetic order, and the assumption here is that these samples are not long range ordered. I think a few edits would be very useful here to make sure that this is conveyed, especially as this article is aimed at a general science journal and the audience will certainly include researchers who regularly look at long range magnetic structures.

Reviewer #3

(Remarks to the Author)

Peng and co-workers describe the application of a multi-technique approach, encompassing conventional SQUID, micro-SQUID, cantilever torque magnetometry, and ¹⁶¹Dy synchrotron Mössbauer spectroscopy, to examine the magnetic transitions in a {CoIII₃DyIII₃} coordination cluster molecule featuring a toroidal moment. The experimental results are in agreement with the results derived from ab initio calculations. Furthermore, the calculation results indicate that a hydrogen-bonded chloride counterion significantly influences the magnetic structure of this cluster. This suggests that optimizing counterions could be a hitherto unexplored chemical approach to tune the toroidal moments. As the authors have noted, this work is the first application of ¹⁶¹Dy synchrotron Mössbauer spectroscopy in the investigation of a polynuclear single-molecule magnet (SMM) system. This would arouse great interest among researchers in the field of SMMs. Overall, the manuscript is well-organized and represents an interesting, high-quality contribution to the field. Therefore, I recommend the manuscript to be published after the following issues have been addressed.

1. Have any coordination compounds similar to [CoIII₃DyIII₃(μ₃-OH)₄(O₂C-C₆H₄-p-Me)₆(p_{mide})₃(H₂O)₃]Cl₂·10MeCN (1) been previously reported, including details of their magnetic properties? In such a case, it is advisable to conduct a comparative analysis of their structures and magnetic properties.
2. The authors should provide the detailed fitting parameters of the Cole-Cole plots, derived from the generalized Debye model, into the supplementary information section.
3. I recommend using “cm⁻¹” as the unit for effective energy barrier (U_{eff}).

4. Full form for the abbreviation "SMTs" (Paragraph 1 on Page 18) should be provided in the manuscript. Furthermore, the authors have missed citing some relevant important references for heterometallic 3d-4f SMTs, such as Nat. Commun., 2017, 8, 1023; Chem. Commun., 2018, 54, 1065; Matter, 2020, 2, 1481, et al.

Version 1:

Reviewer comments:

Reviewer #1

(Remarks to the Author)

The authors have carefully addressed the reviewers' concerns and significantly improved the manuscript. In its current form, the paper meets the journal's standards and can be recommended for acceptance.

Reviewer #2

(Remarks to the Author)

I have now reviewed the manuscript and I believe that the authors have done a good job in addressing my comments.

In particular, I appreciated the following clarifications:

1. The description of what magnetic texture represents.
2. The clarification that the derivative of the magnetisation does display a maximum at higher temperatures. This was an important consideration for me.
3. The description of the measurement sequence over the two beamtimes. With this information, I am satisfied that the sample was immobile.

All other comments have been addressed, and the manuscript is suitable for publication.

This work by Yan Peng, Roberta Sessoli, Volker Schünemann, Annie K. Powell, and their co-workers describes a hexanuclear lanthanide/cobalt complex exhibiting unique magnetic properties. The authors thoroughly investigated the crystal structures and magnetic properties based on both experimental data and computational results. However, the manuscript would benefit from a more discussion on magneto-structural correlations, which would enhance its value for a broader scientific audience. For this reason, the reviewer would recommend this manuscript for publication in Nature Communications after the following major issues are addressed:

1. The authors used $\text{CoCl}_3 \cdot 6\text{H}_2\text{O}$ as the starting material, and low-spin Co(III), which is diamagnetic, was identified in the crystal structure. Consequently, the three Co(III) centers do not contribute to the magnetic properties. Did the authors attempt to use Co(II) instead of Co(III), considering that Co(II) complexes with the same ligand system have demonstrated single-molecule magnet (SMM) behavior?

Answer: As the reviewer points out we have indeed for example previously published a $\text{Co}^{\text{II}}_2\text{Dy}^{\text{III}}_2$ butterfly SMM (<https://doi.org/10.1002/anie.201201478>). However, the objective of the current manuscript was not to produce an SMM but to achieve a toroidal Dy_3 triangle which exhibits exact threefold symmetry and is therefore suitable for single crystal measurements without further possibly interfering paramagnetic 3d ions. In fact, the Co(III) ions provide a framework to enhance the threefold symmetry.

2. Following the previous comment, if the Co(III) centers are diamagnetic, this molecule can be considered a Dy_3 triangular complex with threefold site symmetry. The authors should clarify why this particular molecule is unique and significant for the study of magnetic properties.

Answer: The molecule indeed contains a unique example of a Dy_3 triangle with threefold crystal symmetry imposed by the surrounding diamagnetic Co^{III}_3 triangle. This allows us to study an isolated Dy_3 single molecule toric system in great detail.

3. The checkCIF report provided by the authors does not include the FCF file, and the counterions (Cl^-) are not represented with thermal ellipsoids. Could the authors clarify whether there are any issues with the crystal structure? Please provide an explanation.

Answer: We have now provided a checkCIF that had also checked the structure factors; this only has two extra Alert Gs (referring to missing low-angle reflections blocked by the beamstop) compared to the submitted checkCIF. There are thus no issues with the structure. The hydrogen-bonded chloride was refined anisotropically. The second chloride was disordered against lattice acetonitrile over three crystallographically-equivalent positions, and as might be expected for such disorder in the lattice, anisotropic refinement of the partial-occupancy atoms involved was found to be unsatisfactory. The modelling of this disorder is described both in the CIF under `_refine_special_details` and also in the crystallographic part of the experimental section.

4. In the ^{161}Dy Mössbauer section, the results appear to differ from those reported for mononuclear systems in previous studies. Can one conclude that the magnetic behavior of this multinuclear system is fundamentally different from that of mononuclear complexes?

Answer: Yes, thank you for this remark, this is main message of the paper. Up to a field of 0.6 T the texture coefficient is zero and then it increases in a step like fashion. This clearly show a magnetic transition in the molecule.

Were intramolecular interactions, particularly between Dy–Dy centers, critical to the magnetic properties of this multinuclear compound?

Answer: Yes, as was shown by the detailed *ab initio* calculations. Given the small values of the calculated coupling constants ($\sim 4\text{ cm}^{-1}$), intramolecular interactions are expected to play a significant role only at very low temperatures. For example, the inflection point observed in both computational and experimental studies at low temperatures can likely be attributed to intramolecular coupling. However, the calculated first excited Kramers doublets (KDs) for individual Dy sites (Table S5), as well as the experimentally determined effective energy barrier ($U_{\text{eff}} \approx 29\text{ K}$), are relatively low. These levels may therefore become thermally populated even at modest temperatures, suggesting that at higher temperatures, the magnetic properties are most likely dominated by single dysprosium sites.

Moreover, both the SQUID measurements and computational results suggest the presence of antiferromagnetic interactions. In this research, did spin frustration play a significant role in the formation of the toroidal magnetic state observed in this system?

Answer: The toroidal spin arrangement is in fact an alternative solution to the classical triangular frustration as exemplified by for example Fe_3 triangles. In the paper we refer to several publications which explain the difference between toroidicity and spin frustration (see refs. 6-11) For further information we also refer the reviewer to Coordination Chemistry Reviews 253 (2009) 2328–2341.

5. The authors performed calculations to compare three analogs: $[\text{Dy}_3\text{Co}_3]^{2+}$, $[\text{Dy}_3\text{Co}_3\text{Cl}]^+$, and $[\text{Dy}_3\text{Co}_3\text{Cl}]\text{Cl}$. Indeed, even minor differences—such as changes in molecular charge or the substitution of counterions—can subtly modify the crystal structure and potentially lead to significant changes in magnetic properties. Did the authors observe any structural differences (e.g., bond lengths or angles) in the optimized geometries prior to performing magnetic calculations?

Answer: No structural differences can be observed, as all *ab initio* calculations were performed on the crystal structure in which only the positions of hydrogen atoms were optimized, while the positions of all other atoms were kept fixed, as explained in the ESI.

Moreover, the charge and nature of the counterions have also been shown to play an important role in influencing magnetic properties in recent studies. Could the authors comment on how the magnetic behavior might change if the counterion were substituted with a different monovalent anion?

Answer: This is an interesting question. We did not investigate the effect of other counter anions, but since our calculations show a non-negligible contribution, particularly from the Cl^- anion along the C_3 axis, it can be assumed that the nature of the monovalent anion influences the crystal field around

the Dy³⁺ ions. This, in turn, may induce changes in the magnetic properties. However, without further studies, it is difficult to assess the magnitude of these changes. This effect is not well understood as of yet. We see this as future work with reference to the current system, however bearing in mind that the anion has to conform to the ideal threefold symmetry that we find in the current compound.

Reviewer #2 (Remarks to the Author):

Powell et al report the structure and properties of the magnetotorric compound abbreviated to Co₃Dy₃. They investigate the magnetotorric (MT) properties using a range of techniques including the infrequently reported synchrotron Moessbauer technique.

Overall, this is a very strong piece of work. There are relatively few MT compounds in the literature and even fewer sit on an ideal threefold rotation site in the crystal lattice (and true threefold symmetry is a requirement). As such this is an ideal example to demonstrate the effectiveness of multiple different techniques including conventional magnetometry, Torque magnetometry and CASSCF calculations.

All techniques are well reported and convincingly demonstrate that this is an MT complex. The Dy Moessbauer results are a new technique for this field and given the prevalence of Dy in molecular magnets, this is an interesting technique. I particularly appreciated the clarity of the spectra and how obvious the changes to the spectra are with an applied field. The field dependence of the hyperfine field and texture coefficient are clear, and the latter shows a change which is coincident with that observed by magnetometry. I do have a question regarding this field dependence and equivalence with magnetometry below. If I had to have one criticism of this paper and the technique, it is not clear what additional information is learnt from the Dy Moessbauer. What does the change in magnetic texture represent on the atomic level and what information is encoding in this about the magnetoroidal behaviour that aren't accessible with other techniques.

Answer: The magnetic texture of the magnetic hyperfine field at the ¹⁶¹Dy nucleus models the macroscopic magnetization of the sample. Therefore, the ¹⁶¹Dy nuclei can be viewed as (local) nuclear probes reflecting the samples magnetization. We have stated this now on p. 8.

¹⁶¹Dy SMS rules out the possibility of fast relaxation of the Dy magnetic moments which would lead to zero or small signals in DC and AC-magnetometry experiments.

There are two advantages of the technique: (i) The samples magnetic properties can be sensed from the inside of the molecules -as stated above- and there is no need to disturb the system under study by e.g. magnetic fields (like in magnetometry) or with radiofrequency fields (like in electron paramagnetic resonance). Even more, since the resonant recoil free scattering process of the Mössbauer quanta is a coherent one, the quantum mechanical unavoidable disturbance due to the measuring process is minimized. We have added this statement in the introduction section on p. 3 now in the revised manuscript. (ii) ¹⁶¹Dy SMS has a spectral time window in the ns regime and extends experimental time window of our AC susceptibility measurements which can sense relaxation effects of only down to 10⁻⁵ s.

Overall, I think that this is a rare case of an ideal symmetry MT compound that has been thoroughly studied using a number of techniques. The addition of synchrotron Moessbauer is an excellent addition. With my comments below addressed, this should be published in Nature Communications.

I have a small number of scientific and technical points that I would like the authors to address.

1. The primary identification of the presence of MT is the point of inflection in the magnetisation data. The other experimental and theoretical techniques used verify this. This point of inflection is clear in the dM/dH derivative plotted with the 2 K data (figure 2 inset). However, it is not clear that the same point of inflection is present in the higher temperature M/H data and the derivative of those data is

not presented. This is important as the synchrotron Moessbauer results are presented at 3.6 K and one of the key observations of the data is that the field dependence is the same as the magnetometry at 2 K. However, as the data is presented it is not clear that true MT behaviour is observed at this temperature – my suspicion is that the higher temperature data will have thermal effects due to population of low lying states, so that the M vs H data will not show a point of inflection. This needs to be addressed to make it clear that this is still an MT at 3.6 K. Or that this additional technique is sensitive to MT at these elevated temperatures.

Answer: We thank the referee for these insightful and helpful comments. We indeed believe that the ^{161}Dy SMS technique is sensitive to MT at higher temperatures as the conventional magnetometry given the independence of the Boltzmann population of m_J states as stated in the conclusions. “The magnitude of the hyperfine field is directly related to the magnetic moment of in this case the Dy^{III} ion, however, its magnitude is independent of the Boltzmann population of the m_J states. Nevertheless, using the texture parameter, a Boltzmann population of the m_J states can be modelled, which also gives access to the magnetization of the sample in the ns time window.” Thus, this technique provides an alternative means to identify such transitions at elevated temperatures.

We have provided supplementary data for the derivative plots of the magnetisation at 3 and 5 K. At 3 K we still see a clear maximum at the same field value as for the 2 K data. As the reviewer suggested at 5 K we do see thermal effects. These plots have been added to the ESI.

2. The synchrotron Moessbauer result was done with an unrestrained sample. This is perfectly reasonable as restraining media may give unwanted signals. However, the results are predicated on hoping that the sample was packed tightly enough for reorientation not to occur. I would have liked to see a repeat of the zero-field measurement after the field has been applied to demonstrate the signal was still the same after application of an 4 T field and thus giving some experimental evidence that the sample was stationary.

Answer: In the present work we used the same sample holder and sample preparation procedure as in our work on the mononuclear Dy^{III} single-ion magnet, $[\text{Dy}(\text{Cy}_3\text{PO})_2(\text{H}_2\text{O})_5]\text{Br}_3 \cdot 2 (\text{Cy}_3\text{PO})$. In the latter work we observed reversibility of the magnetic texture of the Dy SMM at 4.2 K in fields up to 4 T. (Ref. 12 of the current Manuscript Scherthan et al. *Angew. Chem. Int. Ed.* 2019 Vol. 58 (11), pp. 3444-3449).

Therefore, for this complex, we did not do the experiment suggested by the reviewer. However, the data presented in Fig. 3b were obtained during two beamtimes. The sample was from the same batch, but freshly prepared into the sample holders at the beamline. During GUP-48137 we measured at 4.2 K $0\text{T} \rightarrow 1\text{T} \rightarrow 2\text{T} \rightarrow 4\text{T}$ and after warming up to room temperature again 0.4T at $T = 4.2\text{K}$. During GUP-55230 we obtained data at $0\text{T} \rightarrow 0.5\text{T} \rightarrow 0.6\text{T} \rightarrow 0.8\text{T}$. Since the data sets are consistent with MT behavior we are confident that the sample was stationary.

3. The only verification of bulk sample purity presented is elemental analysis, while these are within the allowed ranges, I would have hoped to see an X-ray powder diffraction pattern to demonstrate bulk purity.

Answer: We did attempt to obtain a PXRD measurement. However, on measuring the PXRD at room temperature (we don't have the possibility to measure at low temperature), the sample lost solvent

and we couldn't obtain a useful diffraction pattern. We emphasise that the single crystal structure in combination with the C/H/N provides sufficient evidence for the bulk purity.

In addition, I have a comment about some of the language used. The term "spin structure" is used multiple times throughout (including in the abstract where it is referred to as a spin structure that can be correlated with the crystal structure). If I understand the properties of these molecules correctly, these are still fluctuating moments and the spin structures are average ones. Some of the language gets very close to that used to describe magnetic structures with long range magnetic order, and the assumption here is that these samples are not long range ordered. I think a few edits would be very useful here to make sure that this is conveyed, especially as this article is aimed at a general science journal and the audience will certainly include researchers who regularly look at long range magnetic structures.

Answer: The term "spin structure" is commonly used in the molecular magnetism community to describe the relative orientation of anisotropy axes in a molecule. For the general scientific audience, we have added the following clarification in the introduction: "Here spin structure refers to the orientation of the Dy anisotropy axes in one molecular entity"

Reviewer #3 (Remarks to the Author):

Peng and co-workers describe the application of a multi-technique approach, encompassing conventional SQUID, micro-SQUID, cantilever torque magnetometry, and ^{161}Dy synchrotron Mössbauer spectroscopy, to examine the magnetic transitions in a $\{\text{Co}^{\text{III}}\text{Dy}^{\text{III}}\}$ coordination cluster molecule featuring a toroidal moment. The experimental results are in agreement with the results derived from ab initio calculations. Furthermore, the calculation results indicate that a hydrogen-bonded chloride counterion significantly influences the magnetic structure of this cluster. This suggests that optimizing counterions could be a hitherto unexplored chemical approach to tune the toroidal moments. As the authors have noted, this work is the first application of ^{161}Dy synchrotron Mössbauer spectroscopy in the investigation of a polynuclear single-molecule magnet (SMM) system. This would arouse great interest among researchers in the field of SMMs. Overall, the manuscript is well-organized and represents an interesting, high-quality contribution to the field. Therefore, I recommend the manuscript to be published after the following issues have been addressed.

1. Have any coordination compounds similar to $[\text{Co}^{\text{III}}\text{Dy}^{\text{III}}(\mu\text{-OH})_4(\text{O}_2\text{C-C}_6\text{H}_4\text{-p-Me})_6(\text{pmide})_3(\text{H}_2\text{O})_3]\text{Cl}_2 \cdot 10\text{MeCN}$ (1) been previously reported, including details of their magnetic properties? In such a case, it is advisable to conduct a comparative analysis of their structures and magnetic properties.

Answer: There are no other similar $\text{Co}^{\text{III}}\text{Dy}^{\text{III}}_3$ compounds in the literature we can safely compare this compound to. There are other examples of Dy_3 triangles decorated with diamagnetic transition metal ions such as a Zn_3Dy_3 compound (DOI: 10.1002/chem.201703842). However, here the monoclinic crystal symmetry ($P2_1/n$) precludes the oriented single crystal studies reported in the present manuscript where our cluster crystallizes with all of the threefold axes are mutually co-parallel ($P\bar{3}c1$). Furthermore, the bridging modes of the ligands, in particular the central bridging carbonate, used in the Zn-Dy compound are very different to those in our compound and comparison would therefore be uninformative.

2. The authors should provide the detailed fitting parameters of the Cole-Cole plots, derived from the generalized Debye model, into the supplementary information section.

Answer: The fitting parameters are now supplied in the supplementary information (see Table S3).

3. I recommend using "cm⁻¹" as the unit for effective energy barrier (U_{eff}).

Answer: Both units are commonly used in the literature, however for convenience we have added the corresponding values to the main text.

4. Full form for the abbreviation "SMTs" (Paragraph 1 on Page 18) should be provided in the manuscript. Furthermore, the authors have missed citing some relevant important references for heterometallic 3d-4f SMTs, such as Nat. Commun., 2017, 8, 1023; Chem. Commun., 2018, 54, 1065; Matter, 2020, 2, 1481, et al.

Answer: The abbreviation SMT has now been defined in the main text and the suggested references added.